# Autophagy in T cells from aged donors is maintained by spermidine and correlates with function and vaccine responses

Ghada Alsaleh[1†]*, Isabel Panse[1†], Leo Swadling[2], Hanlin Zhang[1], Felix Clemens Richter[1], Alain Meyer[3], Janet Lord[4], Eleanor Barnes[5,6,7], Paul Klenerman[5,6,7], Christopher Green[8], Anna Katharina Simon[1]*

[1]The Kennedy Institute of Rheumatology, NDORMS, University of Oxford, Oxford, United Kingdom; [2]Division of Infection and Immunity, University College London, London, United Kingdom; [3]Fédération de médecine translationnelle Université de Strasbourg, Strasbourg, France; [4]MRC-Versus Arthritis Centre for Musculoskeletal Ageing Research, Institute of Inflammation and Ageing, University of Birmingham, Birmingham, United Kingdom; [5]Peter Medawar Building for Pathogen Research, Nuffield Department of Medicine, University of Oxford, Oxford, United Kingdom; [6]Translational Gastroenterology Unit, John Radcliffe Hospital, Oxford, United Kingdom; [7]NIHR Oxford Biomedical Research Centre, John Radcliffe Hospital, Oxford, United Kingdom; [8]Oxford Vaccine Group, Department of Paediatrics, University of Oxford, Oxford, United Kingdom

*For correspondence:
ghada.alsaleh@kennedy.ox.ac.uk
(GA);
katja.simon@kennedy.ox.ac.uk
(AKS)

[†]These authors contributed equally to this work

Competing interests: The authors declare that no competing interests exist.

**Abstract** Vaccines are powerful tools to develop immune memory to infectious diseases and prevent excess mortality. In older adults, however vaccines are generally less efficacious and the molecular mechanisms that underpin this remain largely unknown. Autophagy, a process known to prevent aging, is critical for the maintenance of immune memory in mice. Here, we show that autophagy is specifically induced in vaccine-induced antigen-specific CD8+ T cells in healthy human volunteers. In addition, reduced IFNγ secretion by RSV-induced T cells in older vaccinees correlates with low autophagy levels. We demonstrate that levels of the endogenous autophagy-inducing metabolite spermidine fall in human T cells with age. Spermidine supplementation in T cells from old donors recovers their autophagy level and function, similar to young donors' cells, in which spermidine biosynthesis has been inhibited. Finally, our data show that endogenous spermidine maintains autophagy via the translation factor eIF5A and transcription factor TFEB. In summary, we have provided evidence for the importance of autophagy in vaccine immunogenicity in older humans and uncovered two novel drug targets that may increase vaccination efficiency in the aging context.

## Introduction

The outbreak of coronavirus disease 2019 (COVID-19) caused a great threat to global public health in 2020 with the majority of deaths occurring in older adults. The development of effective treatments and vaccines against COVID-19 is now more than ever becoming a pressing and urgent challenge to overcome (*Zhang et al., 2020*; *Lurie et al., 2020*). However, the successful vaccination of the elderly against pathogens is considered one of the big challenges in our society (*Weinberger, 2018*; *Chen et al., 2009*). Immunosenescence, which is characterized by poor induction and recall of B and T memory responses upon exposure to new antigens, can lead to reduced immune responses following immunization of older adults. While most vaccines are less immunogenic and

effective in the older population (*Weinberger, 2018*), little is known about the molecular mechanisms that underpin immune senescence. Autophagy is thought to be one of the few cellular processes that underlie many facets of cellular aging including immune senescence (*Zhang et al., 2016*). By delivering unwanted cytoplasmic material to the lysosomes for degradation, autophagy limits mitochondrial dysfunction and accumulation of reactive oxygen species (ROS) (*Rubinsztein et al., 2011*). Autophagy degrades protein aggregates that accumulate with age and its age-related decline could contribute to 'inflamm-aging' (*Salminen et al., 2012*), the age-related increase in inflammatory cytokines in the blood and tissue. Loss of autophagy strongly promotes production of the inflammatory cytokines TNFα, IL-6, and IL1-β (*Saitoh et al., 2008*; *Stranks et al., 2015*). We previously found autophagy levels decline with age in human peripheral CD8$^+$ T cells (*Phadwal et al., 2012*). Deletion of key autophagy genes leads to a prematurely aged immune phenotype, with loss of function in mouse memory CD8$^+$ T cells (*Xu et al., 2014*; *Puleston et al., 2014*), hematopoietic stem cells (*Mortensen et al., 2011*), and macrophages (*Stranks et al., 2015*) with a myeloid bias (*Mortensen et al., 2011*). In addition, we find in autophagy-deficient immune cells the same cellular phenotype that cells display in older organisms; they accumulate ROS and damaged mitochondria (*Stranks et al., 2015*; *Puleston et al., 2014*).

Importantly, we can improve CD8$^+$ T memory responses from aged mice with spermidine (*Puleston et al., 2014*), an endogenous metabolite synthesized from arginine. It was shown in yeast and other model organisms that spermidine extends life-span via increased autophagy (*Madeo et al., 2018*). Several downstream mechanisms of spermidine-induced autophagy have been described in mice, including the inhibition of histone de-acetylases (*Madeo et al., 2018*). Recently, we uncovered a novel pathway in which spermidine donates a residue for the hypusination of the translation factor eIF5a, which is necessary for the translation of a three proline motif present in the master transcription factor of autophagy and lysosomal biogenesis, called TFEB (*Zhang et al., 2019*). We demonstrated that this pathway operates in activated B cells, which upon activation have an unusually high-protein synthesis rate, owing to the high production of immunoglobulins. It is likely that B cells may be particularly reliant on the unfolded protein response, the proteasome, and autophagy, to cope with this high rate of protein synthesis. B cells may have evolved special coping strategies including the translational signaling for autophagy via eIF5A and TFEB. We therefore sought to extend our findings to another immune cell type, CD8+ T cells, to investigate whether this pathway may be conserved in a related adaptive immune subset and possibly broadly applicable.

Here, we show for the first time that autophagy is indeed highly active in human CD8+ T cells after the in vivo encounter of antigens in donors from two different experimental vaccination trials. Our data show that polyamine levels fall with age in peripheral mononuclear cells. When supplemented with spermidine, the dysfunctional autophagic flux can be rejuvenated in CD8+ T cells from old donors, and levels of the important effector molecules IFNγ and perforin are enhanced as a consequence. Moreover, autophagy and effector function are maintained by spermidine in T cells from young donors. Lastly, in human CD8+ T cells we show that spermidine signals via eIF5A and TFEB to maintain autophagy levels. This study demonstrates that the function of human CD8+ T cells can be improved with spermidine. Taken together with our previous work on B cells, this leads us to the hypothesis that both T and B cell responses to infections and vaccinations are exquisitely reliant on sufficient autophagy levels, which are maintained by intracellular spermidine. This work highlights the potential of spermidine as a vaccine adjuvant in the older adults.

## Results

First, we optimized a flow cytometry-based assay to reliably and reproducibly measure autophagy, before applying it to measure autophagy after in vivo antigen-stimulated T cells post-vaccination. To inhibit the autophagic flux and thereby degradation of LC3-II, the lysosomal inhibitor bafilomycin A was added to the culture for 2 hr before washing out non-membrane-bound LC3-I and staining. In Jurkat cells, a human lymphocyte line, we confirmed that treatment with bafilomycin A1 increases LC3-II by both flow cytometry and western blot (*Figure 1—figure supplement 1a and b*). Similar results were obtained in PBMCs stimulated in vitro (*Figure 1—figure supplement 1c and d*). Next, we used in vitro stimulated control PBMCs obtained from five young donors (22–50 years old), bled on three different occasions, 2 weeks apart, were stimulated with either IFNγ/LPS, anti-CD40 /-IgM or anti-CD3/CD28 to induce autophagy in monocytes (*Figure 1—figure supplement 2a*), B cells

(*Figure 1—figure supplement 2b*), CD4+ T (*Figure 1—figure supplement 2c*), or CD8+ T (*Figure 1—figure supplement 2d*) cells, respectively.

Taken together, the data show: (a) autophagic flux varies little within one individual between blood draws; (b) there is limited variation between individuals; (c) bafilomycin A treatment leads to accumulation of LC3-II in all cell types; (d) the respective stimulations via TCR/BCR/TLR or IFNγ-R weakly induce autophagic flux which is consistently further increased in the presence of bafilomycin A.

Published human studies so far have limited their analysis of TCR-induced autophagy to non-specific stimulation in vitro such as anti-CD3/CD28. Here, we took advantage of an existing cohort of healthy human donors that were vaccinated with a candidate HCV vaccine encoding HCV non-structural proteins (NSmut). Healthy volunteers received an adenoviral vector prime vaccination (ChAd3-NSmut or Ad6-NSmut) followed by a heterologous adenoviral boost vaccination (ChAd3-NSmut or Ad6-NSmut) or an MVA-NSmut boost vaccination (*Figure 1a* and *Figure 1—figure supplement 3*; *Swadling et al., 2016*; *Barnes et al., 2012*).

To identify HCV-specific CD8+ T cells, PBMCs were co-stained with an MHC class I pentamer (HLA-A*02:01, HCV peptide KLSGLGINAV) and anti-LC3-II (*Figure 1b*). When measured at the peak magnitude of the T cell response to vaccination (2–4 weeks post Ad, 1 week post MVA) HCV-specific CD8+ T cells show a significant increase in autophagic flux that is not observed in HCV non-specific CD8+ T cells (*Figure 1c*). In antigen-specific T cells, autophagic flux is highest shortly after vaccination but had declined to levels equivalent to antigen-non-specific T cells cells by the end of the study (week 36 or 52). Together, these data show that antigen exposure induces autophagic flux in the antigen-reactive CD8+ T cell subset in humans in vivo. In previous work, we found that autophagic flux was reduced in CD8+ T cells from 24-month-old mice 12. To test whether this is true in human CD8+ T cells from human individuals, and whether this correlates with vaccine immunogenicity, we measured autophagic flux in vaccinees of various ages that were given an experimental RSV vaccine (*Figure 1a* and *Figure 1—figure supplement 4*).

As older adults are particularly susceptible to severe disease from RSV infection, the vaccine was given to two age groups (18–50 years and 60–75 years of age). As expected, the naturally derived population of RSV-specific IFNγ-producing CD8+ T cells in peripheral circulation in response to the infection declines with age (*Figure 1—figure supplement 5*).

No MHC class I pentamers were available for RSV to identify antigen-specific T cells; however, bulk T cells of the >60 year vaccinees showed significantly lower basal autophagic flux (*Figure 1d*). We correlated autophagy levels in T cells with IFNγ ELISpot responses in individual vaccinees and found a strong inverse correlation in the aged group (*Figure 1e*) between autophagy and IFNγ responses, but not in the young group (*Figure 1f*). Taken together these data suggest that reduced T cell autophagy in aged T cells may underpin reduced T cell responses to vaccination. To gain better insight into the role of autophagy in T cells function, we sorted and activated splenocytes from control and autophagy-deficient mice, in which the essential autophagy gene Atg7 was deleted with Cre driven by the CD4 promotor (Atg7$^{\Delta cd4}$). Both intracellular IFNγ expression and secretion was significantly decreased in Atg7-deficient T cells (*Figure 2a and b*). Similarly, intracellular perforin expression was reduced in Atg7-deficient T cells (*Figure 2c*). Next, we sought to confirm our data in human T cells using pharmacological autophagic flux inhibitors, namely hydroxychloroquine (HcQ) that inhibits autophagosomal-lysosomal fusion (*Mauthe et al., 2018*) and the AMPK1 and Ulk1 inhibitor SBI-0206965 (SbI) (*Dite et al., 2018*; *Egan et al., 2015*). Using these drugs, we could recapitulate the data observed in the genetic mouse model of autophagy deletion showing an inhibition of IFNγ (*Figure 2d and e*) and perforin production (*Figure 2f*) in CD8+ T cells. Conversely, activation of autophagy by Resveratrol (Res) (*Rubinsztein et al., 2011*; *Morselli et al., 2010*) in human CD8+ T cell increased the production of IFNγ and perforin (*Figure 2g–i*). In summary, these data suggest that autophagy is required for CD8+ T cell function by controlling IFNγ and perforin production.

We have recently shown that treating old but not young mice with the metabolite spermidine improves autophagy levels in B lymphocytes due to an age-related decline of endogenous spermidine (*Zhang et al., 2019*). Here, we sought to confirm this in human lymphocytes. Thus, we determined spermidine, spermine and putrescine levels in PBMCs by gas chromatography-mass spectrometry (GC-MS), and found an inverse correlation between age and spermidine but not with spermine or putrescine (*Figure 3a*). We hypothesized that low levels of spermidine are responsible for low levels of autophagy and poor T cell function in PBMCs from old donors. We therefore tested

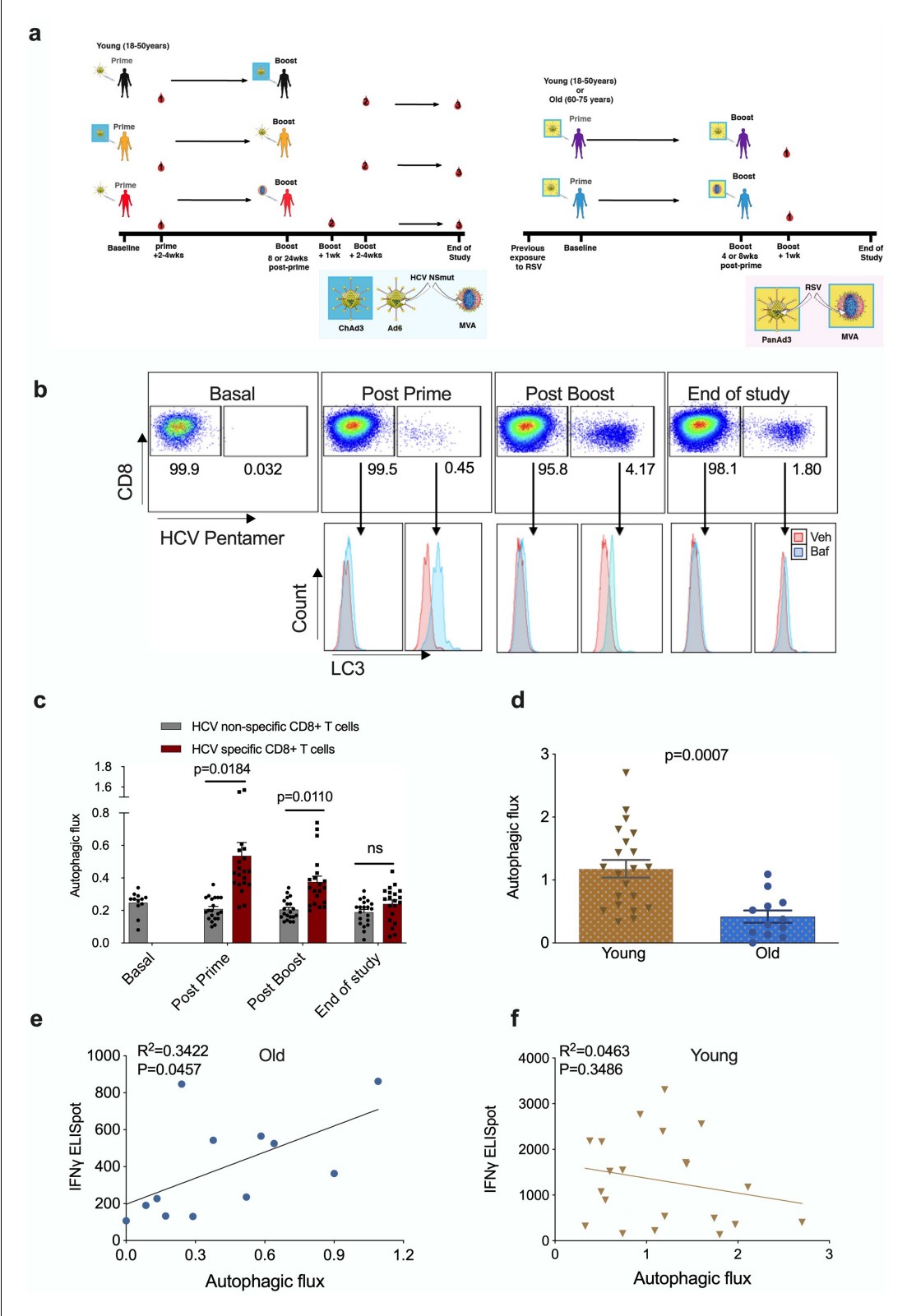

**Figure 1.** Autophagy is induced by vaccination in antigen-specific T cells and correlates with donor age. PBMCs were isolated from blood samples of vaccinated healthy donors. LC3-II was measured in CD8+ cells using flow cytometry after 2 hr treatment with 10 nM bafilomycin A1 (BafA1) or vehicle. Autophagic flux was calculated as LC3-II mean fluorescence intensity (BafA1-Vehicle)/Vehicle. (a) Vaccine regimen for HCV and RSV trials. (b) Representative plots showing BafA1 in light blue and vehicle pink. (c) Quantification in HCV non-specific CD8+ T cells and HCV-specific CD8+ T cells

*Figure 1 continued on next page*

*Figure 1 continued*

detected by HCV pentamers from 10 vaccinees (includling duplicates) using different HCV vaccine regimens, priming with ChimAd and boosting with MVA or AD6 vectors. Autophagy was measured at the peak of the T cell response post prime vaccination, peak of the T cell response post boost vaccination and at the end of the study. (d) Autophagic flux was measured in CD8+ cells from young donors (N = 12,<65 years) and old donors (N = 21, >65 years) vaccinated with respiratory syncytial virus (RSV) 7 days after last boost, quantification calculated as mentioned above. Data represented as mean ± SEM. (e, f) Correlation of autophagic flux with total response to peptide pools specific T cell IFNγ response to RSV exposure measured by ELISpot in CD8+ cells from old donors (e) and young donors (f), donors as in (d). Linear regression with 95% confidence intervals from old and young donors. The goodness of fit was assessed by $R^2$. The p value of the slope is calculated by F test.

The online version of this article includes the following source data and figure supplement(s) for figure 1:

**Source data 1.** Autophagy is induced by vaccination in antigen-specific T cells and correlates with donor age.
**Figure supplement 1.** Autophagy levels by flow cytometry-based assay and conventional LC3 western blot in Jurkat cell line and PBMCs.
**Figure supplement 2.** LC3-II detection by flow cytometry is a reliable and reproducible technique in immune cells over several blood draws.
**Figure supplement 2—source data 1.** LC3-II detection by flow cytometry is a reliable and reproducible technique in immune cells over several blood draws.
**Figure supplement 3.** Regimen of immunizations and blood sampling for HCV trail.
**Figure supplement 4.** Regimen of immunizations and blood sampling for RSV trail.
**Figure supplement 5.** Correlation of age with total and peptide-pool specific T cell IFNγ response to RSV exposure measured by ELISpot in CD8+ cells, donors as in *Figure 1e*.
**Figure supplement 5—source data 1.** Correlation of age with total and peptide-pool specific T cell IFNγ response to RSV.

whether supplementation with spermidine recovers T cell autophagy and function. As activation of PBMCs with anti-CD3/CD28 optimally induces autophagy levels on day 4 (*Watanabe et al., 2014*), we activated PBMCs from old donors in the presence of spermidine for 4 days and tested their autophagic flux and function by flow cytometry. Both autophagic flux and the secretion of IFNγ measured by ELISA was improved significantly in T cells from older vaccinees (*Figure 3b and c*). Similarly, increased IFNγ can be detected after spermidine treatment by intracellular staining for flow cytometry (*Figure 3d*). Interestingly, spermidine supplementation also increases the expression of perforin (*Figure 3e*) but not of Granzyme B (*Figure 3f*). Moreover, to investigate whether spermidine acts through autophagy to improve CD8+ T cell function, we added spermidine together with pharmacological autophagy inhibitors. Indeed, HcQ or Sbl ablates the rejuvenating effects of spermidine supplementation in old donor cells (*Figure 3c–f*). In contrast, spermine, another polyamine from the same pathway, which is not significantly reduced with age (*Figure 3a*) and does not serve as a direct substrate for eIF5A hypusination (*Lee and Park, 2000*), does not rescue CD8+ T cell function from old donors (*Figure 3—figure supplement 1a–c*). These data underline the importance of cellular spermidine levels in aged CD8+ T cells.

Polyamine biosynthesis has previously been shown to increase mitochondrial protein synthesis (*Madeo et al., 2018*; *Puleston et al., 2019*; *Wang et al., 2020*). Moreover, autophagy can ensure mitochondrial quality by selective degradation of damaged or dysfunctional mitochondria, a process called mitophagy, which is critical for maintaining cellular function (*Palikaras et al., 2018*; *Vara-Perez et al., 2019*). Since autophagy levels decline with age and damaged mitochondria accumulate in aged T cells (*Bektas et al., 2019*; *Callender et al., 2020*), we hypothesized that spermidine-induced autophagy may improve mitochondrial function in aged T cells. To test this, we measured mitochondrial function in CD8+ T cells isolated from old donors after spermidine treatment. We indeed show that mitochondrial mass was reduced after spermidine treatment as assessed by Mito-Tracker Green (MTG) and nonylacridine orange (NAO) staining, two fluorescent probes which have been commonly used to assess mitochondrial mass (*Doherty and Perl, 2017*; *Figure 3—figure supplement 2a and b*). The reduction of mitochondrial mass during spermidine treatment could be related to improved clearance of damaged mitochondria by mitophagy or decreased biogenesis. Damaged or dysfunctional mitochondria often show altered membrane potential and increased ROS) production (*Puleston et al., 2014*); however, we did not observe any changes in these parameters after spermidine treatment (*Figure 3—figure supplement 2c and d*). Our data suggest that spermidine treatment lowers mitochondrial content, presumably by mitophagy, but does not improve mitochondrial health. Mitochondria are not the only type of autophagic cargo and other cargos need to be investigated to understand exactly how autophagy induction improves T cell function.

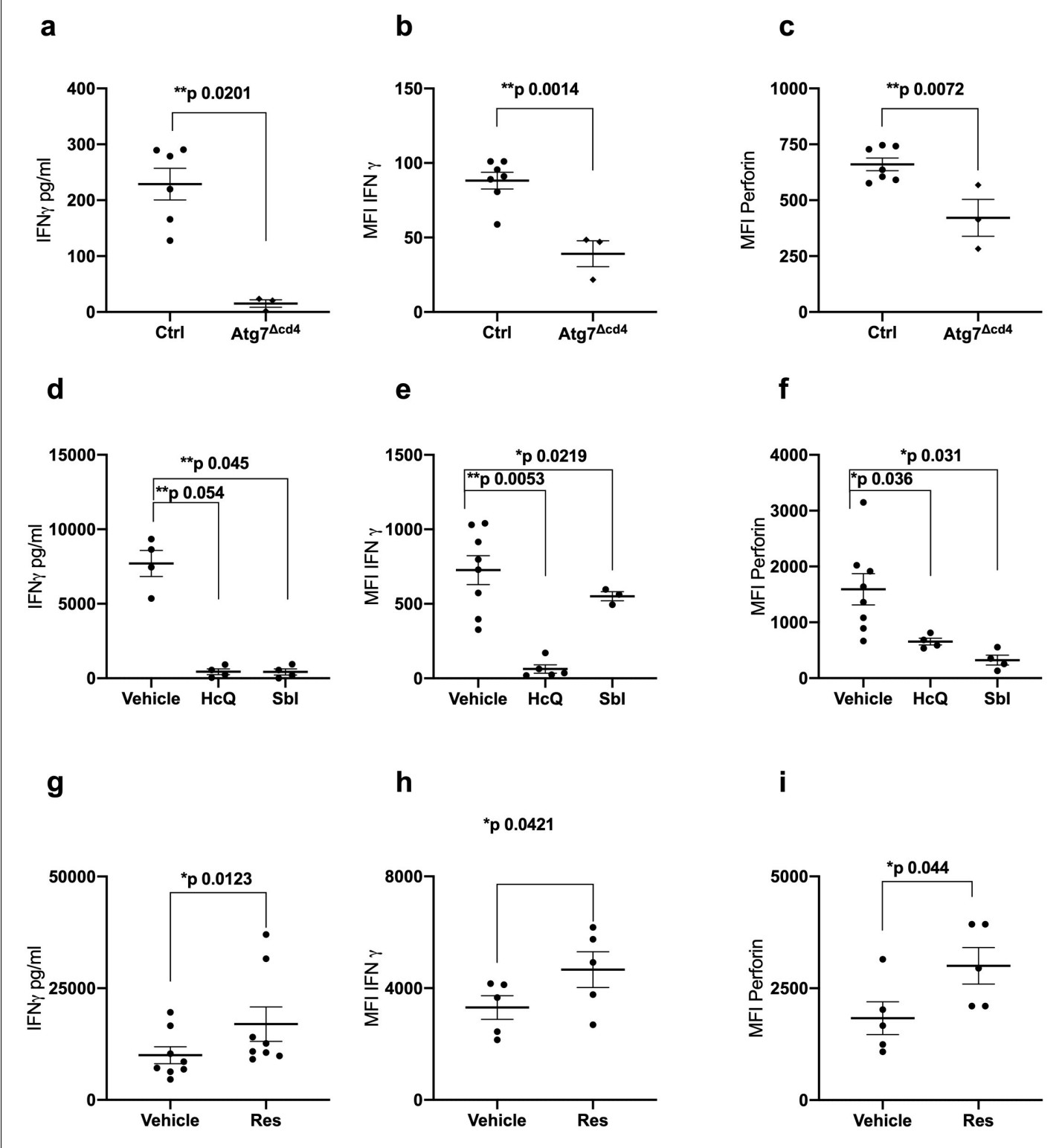

**Figure 2.** Autophagy is required for CD8$^+$ T cell function. (**a–c**) Splenocytes from control mice (Ctrl: CD4-cre;Atg7$^{+/+}$) and Atg7 knockout mice (Atg7$^{\Delta cd4}$: CD4-cre;Atg7$^{-/-}$) were cultured with anti-CD3/CD28 for 4 days and IFNγ assessed by ELISA in tissue culture supernatant (**a**), intracellular IFNγ by flow cytometry (**b**), intracellular perforin by flow cytometry (**c**), all gated on CD8$^+$ T cells. (**d–i**) PBMCs from human donors (>65 years) were cultured with anti-CD3/CD28 for 4 days and where indicated treated with 10 μM Hydroxychloroquine (HcQ), 10 μM BSI-0206965 (SbI), 10 μM Resveratrol (Res) and IFNγ assessed by ELISA in tissue culture supernatant (**d, g**), intracellular IFNγ by flow cytometry (**e, h**), intracellular perforin by flow cytometry (**f, i**), all gated on CD8+ cells. Data represented as mean ± SEM, MFI = mean fluorescence intensity. Statistics by paired t-test for (**d–i**).

*Figure 2 continued on next page*

*Figure 2 continued*

The online version of this article includes the following source data for figure 2:

**Source data 1.** Autophagy is required for CD8$^+$ T cell function.

Next, we investigated whether endogenous polyamine maintains autophagy levels in activated CD8+ T cells from young donors. The drug DFMO (α-difluoromethylornithine) inhibits polyamine biosynthesis by irreversibly inhibiting ornithine decarboxylase (ODC), a key enzyme conversion of ornithine into putrescine. Addition of DFMO almost completely blocked autophagy in CD8+ T cells activated over 7 days;, however, when cells were supplemented with spermidine autophagic flux was recovered (*Figure 4a*). DFMO also partially blocks IFNγ and perforin expression in anti-CD3/CD28 activated CD8+ T cells, which can be rescued by spermidine (*Figure 4b and c*). As before, granzyme B expression was not affected by DFMO or spermidine supplementation (*Figure 4d*). T cells from young donors, with their high endogenous spermidine levels, do not respond to spermidine supplementation (*Figure 4e–g*).

We previously found that spermidine maintains autophagic flux via hypusination of eIF5A which regulates TFEB expression (*Zhang et al., 2019*), and we sought to test this pathway in human T cells. Spermidine provides a moiety to the translation factor eIF5A, which in addition to its role in initiation and termination also promotes the translation of polyproline-rich domains, which are difficult to translate (*Gutierrez et al., 2013*). One such triproline motif-containing protein is TFEB, with mouse TFEB containing one triproline motif while human TFEB contains two. TFEB is the key master transcription factor of autophagosomal and lysosomal gene expression (*Napolitano and Ballabio, 2016*; *Lapierre et al., 2015*; *Settembre et al., 2011*). Here, we addressed whether this pathway operates in human T cells and accounts for the loss of autophagy and T cell function. We first verified in the human lymphocyte line Jurkat that the inhibitor GC7 inhibits the hypusinated/activated form of eIF5A (*Figure 5a*) and also confirmed that it decreases LC3-II expression in a dose-dependent manner (*Figure 5b*). GC7 reduces TFEB and hypusinated eIF5A in CD8+ T cells activated for 4 days (*Figure 5c*). It also diminishes the autophagic flux in activated CD8+ T cells over a time course of 7 days (*Figure 5d*). We then tested whether spermidine maintains eIF5A hypusination and TFEB levels in CD8+ T cells from young donors by depleting endogenous spermidine with DFMO. While anti-CD3/CD28 increases expression levels of TFEB and eIF5A, a reduction of hypusinated eIF5A and TFEB levels in PBMCs treated with DFMO was observed, which is rescued with spermidine (*Figure 5e*). As expected, spermidine treatment of PBMCs from young donors demonstrated to have high levels of spermidine does not induce this pathway (*Figure 5—figure supplement 1*) nor T cell function (*Figure 3e–g*). Finally, we sought to address whether naturally low endogenous spermidine levels can be rescued in PBMCs from old donors activated with anti-CD3/CD28. Again, we observed that activation induces levels of protein expression of both hypusinated eIF5A and TFEB, and spermidine further improves both eIF5A and TFEB two-fold (*Figure 5f*).

To investigate if TFEB is required for T cell function, we knocked out Tfeb in cultured mouse T cells ex vivo (*Figure 5—figure supplement 2a*). Interestingly, both intracellular IFNγ and perforin expression were significantly decreased in Tfeb KO CD8+ T cells upon stimulation (*Figure 5—figure supplement 2b and c*). Moreover, Tfeb KO CD8+ T cells failed to upregulate IFNγ and perforin in the presence of spermidine (*Figure 5—figure supplement 2b and c*). Analogous to previous observation, loss of TFEB had no impact on granzyme B expression (*Figure 5—figure supplement 2d*). Taken together, these data suggest that TFEB itself is essential for T cell function and that spermidine was unable to improve T cell function when TFEB was knocked out.

## Discussion

Several studies have addressed the induction of autophagy by antigenic stimulation in murine T cells in vivo and in response to anti-CD3/CD28 in vitro stimulation of human T cells (*Macian, 2019*). In mouse studies *Xu et al., 2014* measured a peak in autophagy induction shortly after the CD8+ effector phase with subsequent failure to mount memory responses. In line with this, we and Xu et al showed that deletion of autophagy mostly affects T cells in the memory phase of CD8+ T cell responses as opposed to the effector phase (*Xu et al., 2014*; *Puleston et al., 2014*). The question

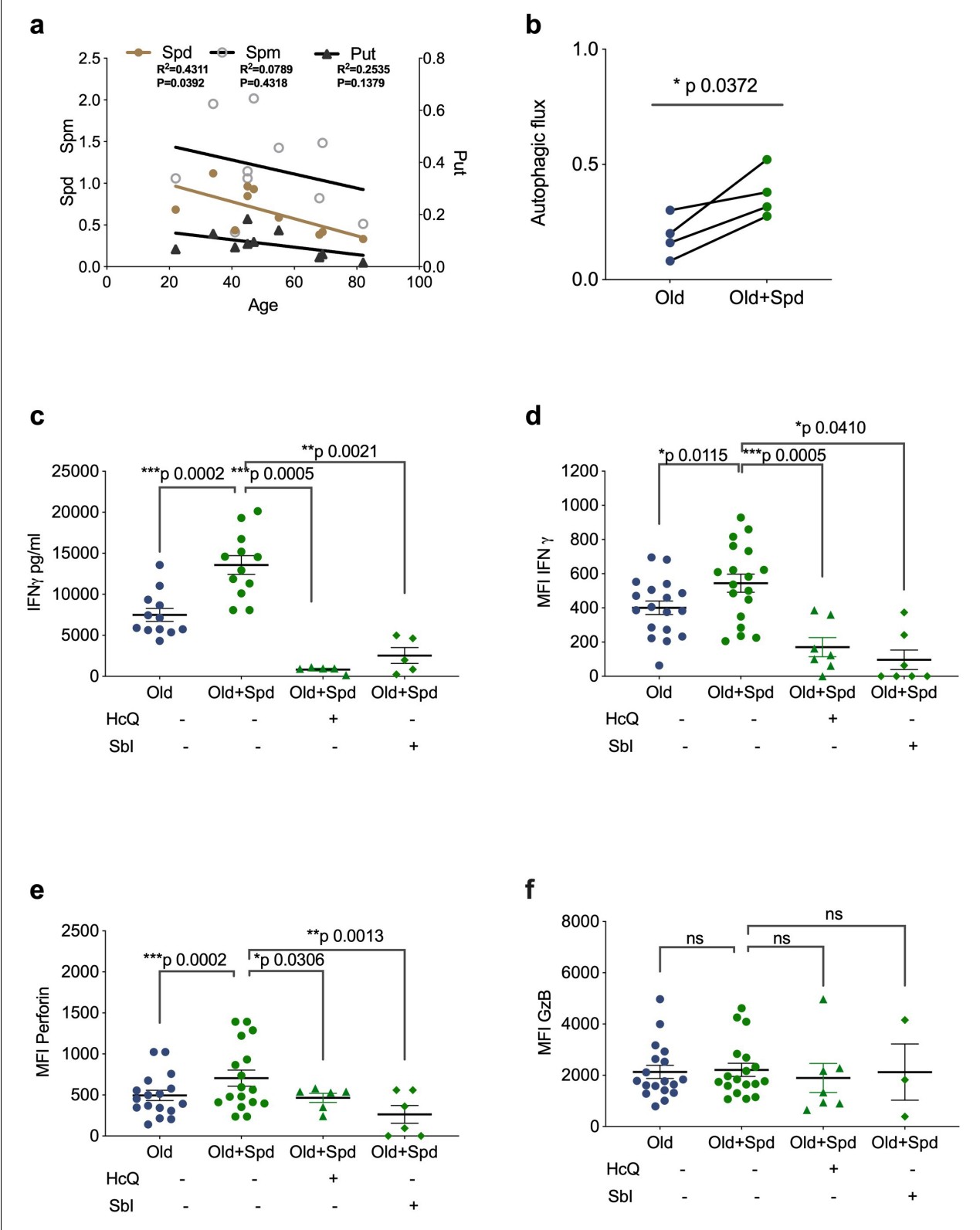

**Figure 3.** Spermidine declines with age and supplementing spermidine improves autophagy and CD8+ T cell function in old donors. (a) Spermidine (Spd), spermine (Spm), and putrescine (Put) content of PBMCs collected from healthy donors were measured by GC-MS. Linear regression with 95% confidence intervals. The goodness of fit was assessed by R (**Lurie et al., 2020**). The p value of the slope is calculated by F test. (b–f) PBMCs from old human donors (>65 years) were cultured with anti-CD3/CD28 for 4 days and where indicated treated with 10 µM spermidine alone or with 10

*Figure 3 continued on next page*

*Figure 3 continued*

µM Hydroxychloroquine (HcQ), 10 µM SBI-0206965 (SbI), and autophagic flux measured by flow cytometry (b), IFNγ assessed by ELISA in tissue culture supernatant (c), intracellular IFNγ by flow cytometry (d), intracellular perforin by flow cytometry (e), intracellular granzyme B (f), all gated on CD8+ cells. Data represented as mean ± SEM, MFI = mean fluorescence intensity. Statistics by paired t-test for (b–f).

The online version of this article includes the following source data and figure supplement(s) for figure 3:

**Source data 1.** Spermidine declines with age and supplementing spermidine improves autophagy and CD8+ T cell function in old donors.
**Figure supplement 1.** Spermine does not improve function of CD8+ T cell from old donors.
**Figure supplement 1—source data 1.** Spermine does not improve function of CD8+ T cell from old donors.
**Figure supplement 2.** Spermidine reduces mitochondrial mass in CD8+ T cell from old donors.
**Figure supplement 2—source data 1.** Spermidine reduces mitochondrial mass in CD8+ T cell from old donors.

remains whether it is initiation or maintenance of memory responses that are affected. Interestingly, the work of *Schlie et al., 2015* showed that the memory response could be rescued by N-acetyl cysteine, arguing that memory cells may be formed but not maintained without autophagy. While it has been shown in human T cells that autophagy peaks after 4 days of anti-CD3/CD28 stimulation, later time points are more difficult to mimic accurately in vitro. Here, we show for the first time that antigenic encounter in vivo induces autophagy after the effector phase, arguing that autophagy may be important in human CD8+ T cells during the memory phase. Interestingly, we could not observe any bystander effect on CD8+ T cells that are not specific for HCV, suggesting that it is a cell-intrinsic effect through specific stimulation via their TCR.

The role of autophagy in these long-term antigen-specific T cells remains still elusive; autophagy could either degrade molecules no longer needed comparable to the transcription factor PU.1 in Th9 cells (*Rivera Vargas et al., 2017*) or cell cycle regulator CDKN1 in the effector phase of CD8+ T cells (*Jia et al., 2015*) or organelles such as mitochondria or ER (*Puleston et al., 2014*; *Jia et al., 2011*). Alternatively, autophagy could provide building blocks including amino acids, free fatty acids and nucleotides that are important for the maintenance of essential cellular processes through provision of ATP (*Shi et al., 2011*; *Pearce et al., 2009*) or transcription and translation for T cell function or a combination of these mechanisms. Our data is in agreement with previous reports showing that spermidine increases the autophagy-dependent selective degradation of mitochondria (mitophagy) and thus contributes to mitochondrial health and functionality (*Eisenberg et al., 2016*; *García-Prat et al., 2016*; *Fan et al., 2017*; *Qi et al., 2016*). The positive impact on cellular mitochondrial content is in line with two recent studies showing that TFEB promotes autophagy and improves mitophagy turnover in neuronal and hepatocytes (*Tan et al., 2019*; *Kim et al., 2018*).

With increasing life expectancy, the number of people over 60 years of age is expected to double by 2050, reaching 2.1 billion worldwide. The severity of many infections is higher in the older population compared to younger adults as particularly notable during the SARS2-CoV pandemic. Moreover, the success of childhood vaccination is widely recognized but the importance of vaccination of the older population is frequently underestimated (*Weinberger, 2018*). Immune responses to vaccines are known to be particularly ineffective in the older population and yet some vaccines such as for influenza and SARS2-Cov are primarily needed for the older adults. The development of drugs that improve vaccination in this expanding population is therefore an urgent socio-economic need.

Targeting mTOR with Rapamycin was the first drug both in mice and in human clinical trials shown to have a beneficial effect on T cell responses in mice and older humans (*Araki et al., 2009*; *Mannick et al., 2018*). However, whether Rapamycin triggers autophagy at the administered dose has not been investigated. With this study we show for the first time that the autophagy-inducing drug spermidine has an immune boosting effect on the T cell compartment in humans in vitro and that low autophagy levels correlate with low responses to vaccination. The translation of TFEB is one of the limiting factors for sufficient autophagy levels required to mount an immune response. In addition, to be active, TFEB needs to be dephosphorylated for its translocation into the nucleus, which mTOR inhibition facilitates (*Martina et al., 2012*; *Roczniak-Ferguson et al., 2012*; *Settembre et al., 2012*). Therefore, spermidine and an mTOR inhibitor may have to be combined to optimally restore immune responses in the older adults.

We find that spermidine levels decline in PBMCs in the older humans. This confirms earlier study in plasma in which spermidine levels were found to be low in the >65 age group and rising again in

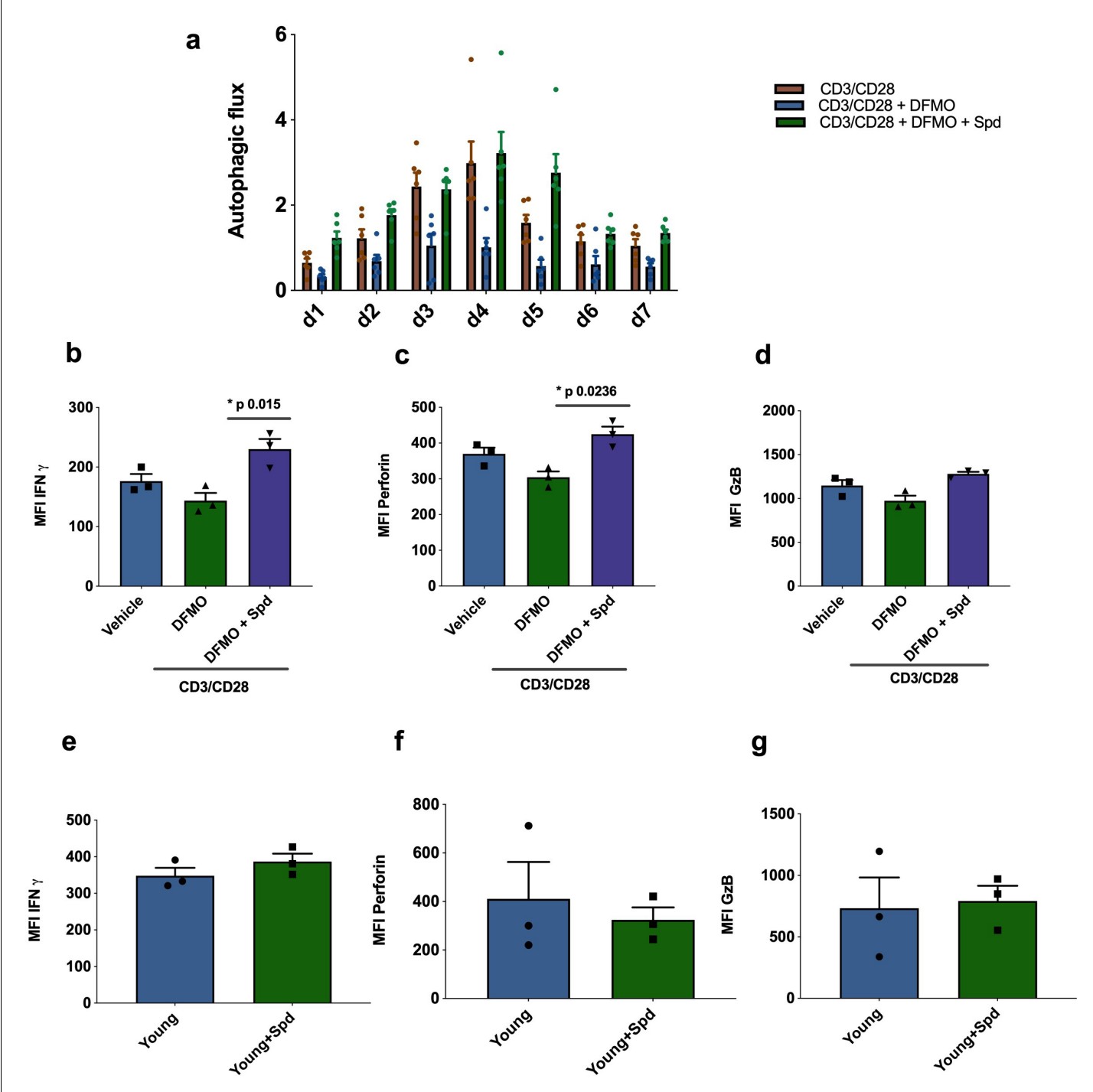

**Figure 4.** Endogenous spermidine maintains levels of autophagy and T cell function. (**a–d**) PBMCs cells from young human donors (<65 years) were activated with anti-CD3/CD28 for 7 days and treated with spermidine synthesis inhibitor 1 mM DFMO alone or together with 10 μM spermidine (Spd). Autophagic flux (**a**) was assessed each day and IFNγ (**b**), Perforin (**c**), Granzyme B (**d**) were measured by flow cytometry in CD8+ cells on day 4. (**e–f**) PBMCs cells from young human donors (<65 years) were cultured with anti-CD3/CD28 for 4 days and streated with 10 μM spermidine. (**e**) Intracellular IFNγ, (**f**) intracellular perforin, (**g**) and intracellular granzyme B were measured in CD8+ cells by flow cytometry. Data represented as mean ± SEM. The online version of this article includes the following source data for figure 4:

**Source data 1.** Endogenous spermidine maintains levels of autophagy and T cell function.

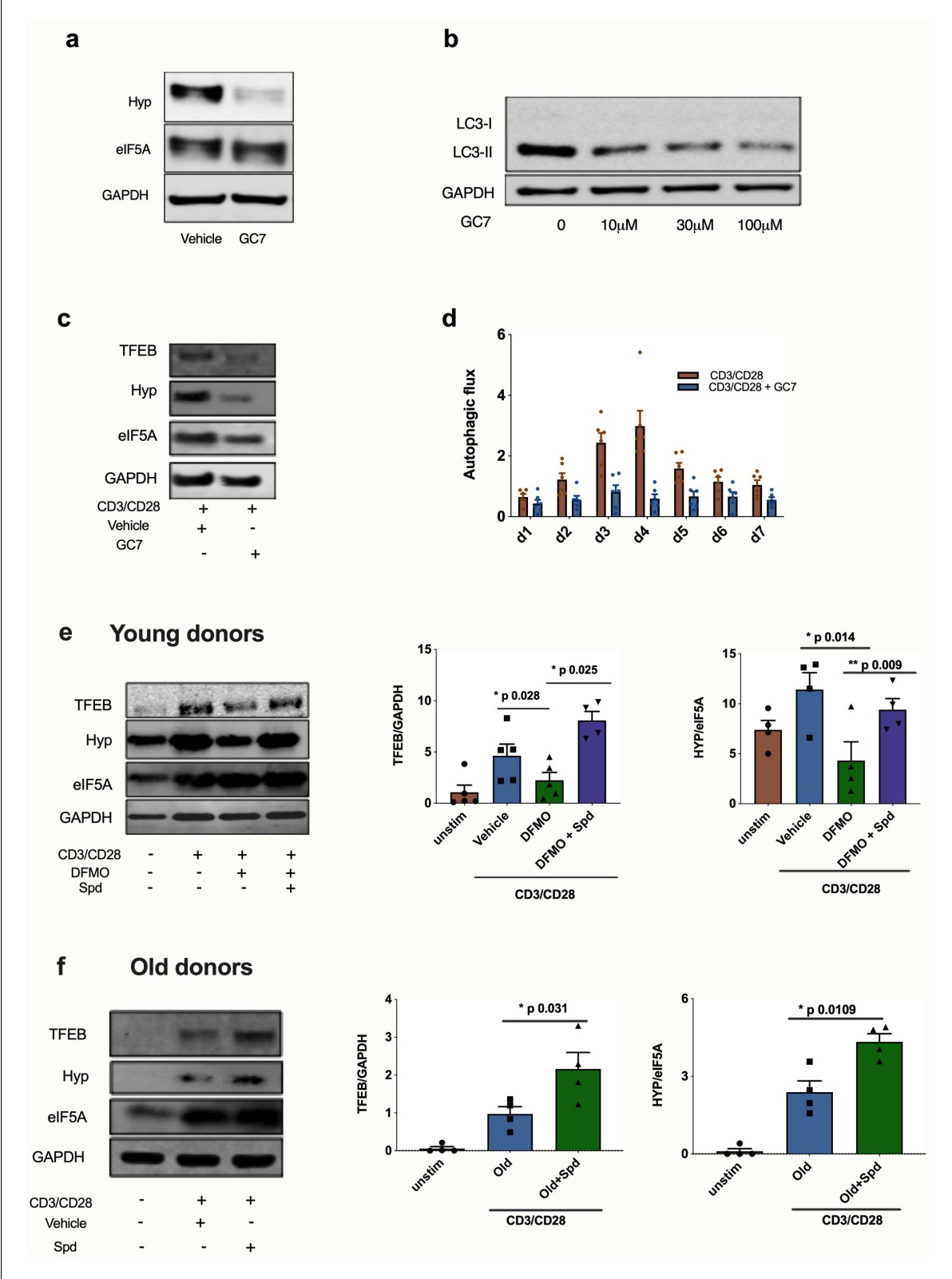

**Figure 5.** Spermidine's mode of action is via eIF5A and TFEB in human CD8+ T cells. (a–d) Human T cell line Jurkat was cultured for 24 hr with 100 μM GC7, then eIF5A and hypusinated eIF5A were measured by WB (a). Jurkat cell line was stimulated with increasing concentrations of GC7 and cell lysates blotted for LC3B (b). (c–d) PBMCs from young human donors were cultured with anti-CD3/CD28 for 7 days and treated with GC7. The protein levels of TFEB and eIF5A hypusination were measured in CD8+ cells by Western blot on day 4 (c) and autophagic flux was determined as in *Figure 1*

*Figure 5 continued on next page*

*Figure 5 continued*

(**d**). PBMCs from young human donors were cultured with anti-CD3/CD28 for 4 days and treated with spermidine synthesis inhibitor DFMO alone or together with 10 µM spermidine and the protein levels of TFEB and eIF5A hypusination were measured in CD8+ cells by wWestern blot (**e**), representative images (left) and quantified (right). PBMCs from old human donors (>65 years) were cultured with anti-CD3/CD28 for 4 days and where indicated treated with 10 µM spermidine and the protein levels of TFEB and eIF5A hypusination were measured in CD8+ cells by wWestern blot (**f**), representative images (left) and quantified (right). Target band intensity was normalized to eIF5A (for Hyp) or GAPDH (for TFEB). Data represented as mean ± SEM.

The online version of this article includes the following source data and figure supplement(s) for figure 5:

**Source data 1.** Dpermidine's mode of action is via eIF5A and TFEB in human CD8+ T cells.
**Figure supplement 1.** Spermidine does not improve eIF5A and TFEB in young donors.
**Figure supplement 1—source data 1.** Spermidine does not improve eIF5A and TFEB in young donors.
**Figure supplement 2.** TFEB is required for CD8+ T cell function.
**Figure supplement 2—source data 1.** TFEB is required for CD8+ T cell function.

centenarians (*Pucciarelli et al., 2012*). Studies of this kind usually indicate that the phenotype is maladaptive with age as centenarians do not display many of the aging features of the age group below. However, the reasons for the age-related decline in spermidine in blood cells are not clear, and currently under investigation. Overall it is evident that endogenous spermidine maintains autophagy in human T cells, a novel metabolic pathway that has not been investigated before.

Spermidine has recently been administered to humans in a small experimental trial with a beneficial effect on cognitive function without adverse effects (*Schwarz et al., 2018*; *Wirth et al., 2019*). It remains to be shown whether at such low doses it has an effect on autophagy. Our study provides a good starting point for a small experimental trial to assess whether spermidine can be used to improve vaccination efficiency in older adults. Furthermore, this trial could help to determine if eIF5A, TFEB and autophagy in PBMCs could be used as biomarkers for the antiaging effect of spermidine and related drugs during vaccination. Based on this information, the vaccination protocol in older adults could in the future be adjusted according to whether this pathway is functional in individual vaccinees.

In conclusion, we validated a novel anti-immune senescence pathway in humans with druggable targets and biomarkers, which could also be used for other broader anti-aging drug trials.

# Materials and methods

## Key resources table

| Reagent type (species) or resource | Designation | Source or reference | Identifiers | Additional information |
|---|---|---|---|---|
| Genetic reagent (*M. musculus*) | Atg7$^{flox}$ | PMID:15866887 | MGI:3587769 | Dr. Masaaki Komatsu (Juntendo University) |
| Genetic reagent (*M. musculus*) | TFEB$^{flox}$ | PMID:23604321 | | Prof. Andrea Ballabio (Telethon Institute of Genetics and Medicine (TIGEM)). |
| Cell line (*H. sapiens*) | Jurkat E6.1 | ATCC | RRID:CVCL_0367 | |
| Antibody | Rabbit anti-GAPDH | Millipore | MAB374, RRID:AB_2107445 | WB (1:5,000) |
| Antibody | Rabbit anti-LC3 | Sigma | L8918, RRID:AB_1079382 | WB (1:1500) |
| Antibody | Mouse anti-eIFA5 | BD Biosciences | 611976, RRID:AB_399397 | WB (1:3000) |
| Antibody | Rabbit anti-hypusine | Millipore | ABS1064, RRID:AB_2631138 | WB (1:1500) |
| Antibody | Rabbit anti-TFEB | Bethyl | A303-673A, RID:AB_11204751 | WB (1:1500) |

*Continued on next page*

*Continued*

| Reagent type (species) or resource | Designation | Source or reference | Identifiers | Additional information |
|---|---|---|---|---|
| Antibody | Mouse anti-Actin | Cell Signaling | 3700, RRID:AB_2242334 | WB (1:10,000) |
| Antibody | IRDye 800CW Donkey Anti-Rabbit IgG (H+L) | LI-COR | 926–32213, RRID:AB_621848 | WB (1:10,000) |
| Antibody | IRDye 680RD Donkey Anti-Mouse IgG (H+L) | LI-COR | 926–68022, RRID:AB_10715072 | WB (1:10,000) |
| Antibody | BV605 anti-human CD14 | BioLegend | 301834, Clone: M5E2 | FACS (1:200) |
| Antibody | PE anti-human Granzyme B | eBioscience | 12-8899-41, Clone: GB11 | FACS (1:100) |
| Antibody | PE/Cy7 anti-human CD8a | BioLegend | 344712, Clone: SK1 | FACS (1:100) |
| Antibody | PE/Cy5 anti-human CD19 | BioLegend | 302210, Clone:HIB19 | FACS (1:100) |
| Antibody | BV711anti-human CD19 | BioLegend | 302245, Clone:HIB19 | FACS (1:200) |
| Antibody | APC anti-human CD3 | BioLegend | 300312, Clone:HIT3a | FACS (1:200) |
| Antibody | APC anti-human Perforin | BioLegend | 353311, Clone:B-D48 | FACS (1:100) |
| Antibody | APC anti-human CD4 | BioLegend | 317416, Clone:OKT4 | FACS (1:200) |
| Antibody | Alexa Fluor 700 anti-human IFNγ | BioLegend | 506515, Clone:B27 | FACS (1:100) |
| Antibody | PE/Cy7 anti-mouse CD8a Antibody | BioLegend | 100722, RRID:AB_312761 | FACS (1:200) |
| Antibody | BV605 anti-mouse CD4 Antibody | BioLegend | 100451, Clone: GK1.5 | FACS (1:200) |
| Antibody | Alexa Fluor 700 anti-mouse IFNγ | BioLegend | 505823, Clone: XMG1.2 | FACS (1:100) |
| Antibody | APC anti-mouse Perforin | BioLegend | 154303, Clone: S16009A | FACS (1:100) |
| Antibody | PE anti-mouse Granzyme B | BioLegend | 372207, Clone: QA16A02 | FACS (1:100) |
| Antibody | APC anti-human CD19 | BioLegend | 302212, RRID:AB_314242 | FACS (1:200) |
| Antibody | Alexa Fluor 700 anti-mouse CD8a | BioLegend | 100730, RRID:AB_493703 | FACS (1:100) |
| Antibody | PE-labeled MHC class I pentamer (HLA-A*02:01, HCV peptide KLSGLGINAV) | ProImmune | (HLA-A*02:01, HCV peptide KLSGLGINAV) | FACS (1:50) |
| Commercial assay or kit | FlowCellect Autophagy LC3 Antibody-based Detection Kit | Merck Millipore | FCCH100171 | FACS (1:20) |
| Commercial assay or kit | LIVE/DEAD Fixable Near-IR Dead Cell Stain Kit | Invitrogen ThermoFisher | L10119 | FACS (1:1000) |
| Commercial assay or kit | LIVE/DEAD Fixable Aqua Dead Cell Stain Kit | Invitrogen ThermoFisher | L34957 | FACS (1:1000) |
| Commercial assay or kit | CellTrace Violet Cell Proliferation Kit | Invitrogen, ThermoFisher | C34557 | FACS (1:1000) |
| Commercial assay or kit | MitoTracker Green | Invitrogen, ThermoFisher | M7514 | FACS (150 nM) |
| Commercial assay or kit | MitoSox | Invitrogen, ThermoFisher | M36008 | FACS (5 uM) |
| Commercial assay or kit | CD3/CD28 activation Dynabeads | Thermo Fisher | 11161D | |
| Commercial assay or kit | Human IFN-γ ELISpot$^{PLUS}$ kit (ALP) | Mabtech | 3420-4APT-2 | |
| Commercial assay or kit | IFNγ ELISA Kit | Life Technologies Ltd | 88-7316-22 | |
| Commercial assay or kit | EasySep Human CD8+ T Cell Isolation Kit | Stemcell | 17953 | |

*Continued on next page*

*Continued*

| Reagent type (species) or resource | Designation | Source or reference | Identifiers | Additional information |
|---|---|---|---|---|
| Commercial assay or kit | BCA Assay | Thermo Fisher | 23227 | 100 µl/sample |
| Chemical compound, drug | Spermidine | Cayman Chemical | 14918 | 10 µM |
| Chemical compound, drug | Spermine | Cayman Chemical | 18041 | 10 µM |
| Chemical compound, drug | Hydroxychloroquine Sulfate | Stratech Scientific Ltd | B4874-APE-10mM | 10 µM |
| Chemical compound, drug | Bafilomycin A1 | Cayman Chemical | 11038 | 10 µM |
| Chemical compound, drug | Resveratrol | Stratech Scientific Ltd | A4182-APE | 10 µM |
| Chemical compound, drug | BSI-0206965 | Stratech Scientific Ltd | A8715-APE | 10 µM |
| Chemical compound, drug | AICR | Stratech Scientific Ltd | A8184-APE | 10 µM |
| Chemical compound, drug | NAO | Thermo Fisher Scientific | A1372 | 100 nM |
| Chemical compound, drug | TRMR | Thermo Fisher Scientific | T668 | FACS (100 nM) |
| Chemical compound, drug | Difluoromethylornithine (DFMO) | Enzo Life Sciences | ALX-270–283 M010 | 1 mM |
| Chemical compound, drug | GC7 | Millipore | 259545–10 MG | 10 µM |
| Chemical compound, drug | 4-Hydroxytamoxifen | Sigma | H7904 | 100 nM |
| Chemical compound, drug | LPS | Santa Cruz | sc-3535 | 10 µg/ml |
| Chemical compound, drug | Aanti-IgM | Jackson Immuno Research | 109-005-043 | 5 µg/ml |
| Chemical compound, drug | Aanti-CD40L | Enzo Life science | ALX-522–015 C010 | 100 ng/ml, |
| Peptide, recombinant protein | Human IFNγ | Enzo Life science | ENZ-PRT141-0100 | 20 ng/ml |
| Software, algorithm | Image Studio Lite | LI-COR | | |
| Software, algorithm | Prism | GraphPad | | |

## Human samples

Human peripheral blood mononuclear cells (PBMCs) were obtained under the ethics reference NRES Berkshire 13/SC/0023, from phase I clinical trials of novel viral-vectored vaccines for hepatitis-C virus (HCV; NCT0100407 and NCT01296451) or respiratory syncytial virus (RSV), described in more detail elsewhere (*Swadling et al., 2016*; *Barnes et al., 2012*; *Green et al., 2015a*; *Green et al., 2015b*; *Green et al., 2019*). Volunteers were self-selected adults who provided written informed consent and who were carefully screened for being healthy before vaccination. The vaccine schedules are described in Diagrams (*Figure 1—figure supplements 3* and *4*). Blood samples were collected in heparinized tubes for assays that required PBMCs. PBMCs were isolated within 6 hr of sample collection. An aliquot of PBMCs was immediately used for fresh ELISpot assays and the remainder cryopreserved in RecoveryTM Cell Freezing Medium. Serum samples were obtained by centrifugation of whole blood collected in clotted tubes, and then cryopreserved.

## Control PBMCs

Control PDBMCs were isolated from blood or blood cones of healthy donors using Ficoll-Paque density gradient separation. All volunteers provided written informed consent. The study was approved by the Local Ethics Committee Oxford and Birmingham. Freshly isolated PBMCs were cultured directly or were frozen in 90% FBS and 10% DMSO in liquid nitrogen. Fresh or thawed PBMCs were cultured with RPMI 1640, 10% FCS, 2 mM L-Glutamine, 100 U/mL Penicillin (Invitrogen).

## ELISpot assay

Ex vivo IFNγ ELISpot assays were performed according to manufacturers' instructions (Mabtech) on freshly isolated PBMCs plated in triplicates at $2 \times 10^5$ PBMCs per well. In brief, peptide pools consisted of mainly 15-mer sequences with 11 amino acid overlaps and covering the sequence of proteins F, N, and M2- 1. Peptides were dissolved in 100% DMSO and arranged in four pools. DMSO (the peptide diluent) and Concanavalin A (ConA) were used as negative and positive controls, respectively. The mean + 4 StDev of the DMSO response from all samples identified a cut off whereby individual samples with background DMSO values $\geq$ 50 spot forming cells per million PBMCs were excluded from analysis. Calculation of triplicate well variance was performed as described previously (*Green et al., 2019*).

## Human T cell assays

The mean age of young donors was 40.7 $\pm$ 11.3 years and the mean age of old donors was 77.6 $\pm$ 6.6. PBMCs were activated with either soluble anti-CD3 (1 µg/mL, Jackson Immuno Research) and anti-CD28 (1 µg/mL, Jackson Immuno Research) with or without 10 µM spermidine, 10 µM spermine or 1 mM difluoromethylornithine (DFMO, Enzo Life Sciences) or 10 µM GC7 (or as indicated, Millipore), 10 µM hydroxychloroquine (Stratech Scientific Ltd), 10 µM SBI-0206965 (Stratech Scientific Ltd) 10 µM Resveratrol (Stratech Scientific Ltd) for 4 days. After MACS sorting of CD8+ T cell using a negative selection kit (CD8+ T Cell Isolation Kit II, human, Miltenyi Biotec), cells were lysed for western blotting or stained for Autophagy flux assay as described below. IFNγ release in culture supernatants was measured by heterologous two-site sandwich ELISA, according to the manufacturer's protocol (Invitrogen).

## Cell line

Human Jurkat T cell line was cultured with increasing concentrations of GC7 (10 µM, 30 µM, 100 µM, Millipore) for 24 hr and then treated with or without Bafilomycin A1 for the last 2 hr. The cell line was mycoplasma negative.

## Mice T cell assays

Splenocytes from: CD4$^{-cre}$;Atg7$^{-/-}$ (Atg7$^{\Delta cd4}$) or CD4$^{-cre}$;Atg7$^{+/+}$ mice were cultured with anti-CD3/CD28 for 4 days. For 4-Hydroxytamoxifen (4-OHT) 4-OHT-inducible Tfeb knockout splenic T cells (CAG-Cre/Esr1+,Tfebf/f), cells were stimulated were anti-CD3/CD28 and 100 nM 4-Hydroxytamoxifen (4-OHT) for 4 days. Animal experiments were approved by the local ethical review committee and performed under UK project licenses PPL 30/3388.

## Autophagy flux assay for flow cytometry

PBMCs from healthy donors, activate for T cells with CD3/CD28 beads (Dynabeads Thermo Fisher) (1:1), for B cells with anti-IgM (5 µg/mL, Jackson Immuno Research) and CD40L (100 ng/mL, Enzo Life science) and for monocytes activated with IFNγ (20 ng/mL, Enzo Life science) and/or LPS (10 µg/mL, Santa Cruz) (all 24 hr except for T cells which were stimulated overnight). Autophagy levels were measured after 2 hr treatment with bafilomycin A1 (10 nM BafA1) or vehicle. We adapted the Flow-Cellect Autophagy LC3 antibody-based assay kit (FCCH100171, Millipore) as follows: In brief, cells were stained with surface markers (as above) and washed with Assay Buffer in 96 well U bottom plates. 0.05% Saponin was added to each well and spun immediately, followed by anti-LC3 (FITC) at 1:20 in Assay Buffer, (30–50 µL/ well) at 4°C for 30 min, and washed once with 150 µL Assay Buffer. Stained cells were fixed with 2% PFA before FACS analysis. Autophagic flux was calculated as LC3-II mean fluorescence intensity of (BafA1-Vehicle)/Vehicle.

## Surface staining for flow cytometry

For CellTrace staining, CellTrace Violet (C34557, Thermo Fisher) was used according to the manufacturer's protocol. Cells were transferred to a round bottom 96 well plate and centrifuged (300xg, 5 min). The pellet was resuspended in PBS containing the viability dye Live/Dead (Life Technologies) or fixable Zombie Aqua Live/Dead (423102, Biolegend) for 10 min in the dark at room temperature (RT) to exclude dead cells during analysis. After washing with PBS/5% FCS, cells were resuspended in PBS/2% FCS/5 mM EDTA (FACS buffer) containing a cocktail of antibodies relevant to the desired cell surface markers. Fc block was typically added to the antibody mix to minimize non-specific staining. Surface antibody staining was performed at 4°C for 20 mins in the dark. A list of all surface antibodies utilized and their working concentrations are in Key resources table. Following incubation cells were washed with FACS buffer and immediately analyzed on a four-laser LSR Fortessa X-20 flow cytometer. Acquired data were analyzed using FlowJo 10.2.

Mitochondrial staining was performed after Live/Dead and surface marker staining by incubating cells at 37°C for 15 min with 5 µM MitoSOX red, 37°C for 25 min with 150 nM MitoTracker Green, 37°C for 25 min with 100 nM nonylacridine orange (NAO) or at 37°C for 15 min with 100 nM tetramethylrhodamine methyl ester (TMRM) (all from Thermo Fisher Scientific).

## Intracellular staining for flow cytometry

For intracellular staining, PBMCs were stimulated in R10 with anti-CD3 (1 µg/ml, Jackson Immuno Research) and anti-CD28 (1 µg/mL, Jackson Immuno Research) with or without 10 µM spermidine for 4 days. On day 4, cells were re-stimulated with the same concentrations of anti-CD3/CD28 for 6 hr at 37°C in the presence of 1 µg/mL brefeldin-A (Sigma-Aldrich). As a control, cells were left unstimulated. Following surface marker staining as described above, cells were fixed with 100 µL Fixation buffer (eBioscience) for 20 min at RT in the dark. Next, cells were permeabilized with 100 µL of 1x Permeabilization buffer (eBioscience) for 15 min in the dark at RT. Then, cells were resuspended in the intracellular antibody mix (anti-IFN-γ, anti-Granzyme B, anti-Perforin) and incubated for 30 min in the dark at RT. After being washed twice with Permeabilization buffer the cells were resuspended in 200 µL of FACS buffer for analysis.

## MHC class I pentamer staining to identify antigen-specific T cells

An HCV-specific HLA-A*02-restricted pentamer, peptide sequence KLSGLGINAV (Proimmune) was used to identify HCV-specific CD8+ T cells ex vivo. PBMCs were washed in PBS and were stained with pentamers at room temperature (20 min) in PBS, washed twice in PBS before further mAb staining as described above. During analysis, stringent gating criteria were applied with doublet and dead cell exclusion to minimize nonspecific binding contamination.

## Western blot

Cells in suspension were washed with PBS and lysed using NP-40 lysis buffer containing proteinase inhibitors (Sigma) and phosphatase inhibitors (Sigma) on ice. After spinning down the debris, protein concentration in the supernatant was quantified by BCA Assay (23227, Thermo Fisher). Reducing Laemmli Sample Buffer was then added to the supernatant and heated at 100°C for 5 min. 5–20 µg protein was loaded for SDS-PAGE analysis. NuPAGE Novex 4–12% Bis-Tris gradient gel (Thermo Fisher) with MES running buffer (Thermo Fisher) was used. Proteins were transferred to a PVDF membrane (IPFL00010, Millipore) and blocked with 5% skimmed milk-TBST. Membranes were incubated with primary antibodies dissolved in 1% milk overnight and secondary antibodies dissolved in 1% milk with 0.01% SDS for imaging using the Odyssey CLx Imaging System. Data were analyzed using Image Studio Lite.

## Spermidine measurement in cell lysates by GC-MS

This protocol was used as published previously (*Yu et al., 2017*). Briefly, cells were washed with PBS and the pellet resuspended in lysis buffer (80% methanol + 5% Trifluoroacetic acid) spiked with 2.5 µM 1,7-diaminoheptane (Sigma). The cell suspension, together with acid-washed glass beads (G8772, Sigma), was transferred to a bead beater tube and homogenized in a bead beater (Precellys 24, Bertin Technologies) for four cycles (6500 Hz, 45 s) with 1 min of ice incubation between each cycle. The homogenized samples were centrifuged at 13,000 g for 20 min at 4°C. The supernatant

was collected and dried overnight. For chemical derivatization, 200 μL trifluoroacetic anhydride was added to the dried pellet and incubated at 60°C for 1 hr, shaking at 1200 rpm. The derivatization product was dried, re-suspended in 30 μL isopropanol and transferred to glass loading-vials. The samples were analyzed using a GCxGC-MS system as described (*Yu et al., 2017*). The following parameters were used for quantification of the 1D-GC-qMS data: Slope: 1000/min, width: 0.04 s, drift 0/min and T. DBL: 1000 min without any smoothing methods used.

## Statistical analyses

Prism software (GraphPad) was used for statistical analyses. Data are represented as mean ± SEM. All comparative statistics were post-hoc analyses. Paired or unpaired two-tailed Student's t-test was used for comparisons between two normally distributed data sets with equal variances. Linear regression with a 95% confidence interval was used to assess the relationships between age and the expression of target proteins or spermidine levels, in which R (*Lurie et al., 2020*) was used to assess the goodness of fit and the p value calculated from F test was used to assess if the slope was significantly non-zero. p Values were used to quantify the statistical significance of the null hypothesis testing. *$p \leq 0.05$, **$p \leq 0.01$, ***$p \leq 0.001$, ****$p \leq 0.0001$, ns, not significant.

## Acknowledgements

We acknowledge Reithera/Okairos for RSV trial, Zhanru Yu for Spermidine and Putrescine measurements, Kirsty McGee and Christos Ermogenous for ethical approval and provision of samples from Birmingham healthy older donors. We thank all donors, volunteers and phlebotomists. K.S. is a Wellcome Investigator, which funded this study. Wellcome funding (WT109665MA) and NIHR SF was also awarded to P.K. We acknowledge funding from the National Institute for Health Research (NIHR) Oxford Biomedical Research Centre (BRC). The views expressed are those of the authors and not necessarily those of the NHS, the NIHR or the Department of Health. Tfeb^flox mice were obtained from A Ballabio.

## Additional information

### Funding

| Funder | Grant reference number | Author |
| --- | --- | --- |
| Wellcome | WT109665MA | Anna Katharina Simon |
| National Institute for Health Research | | Paul Klenerman |
| Oxford Biomedical Research Center | | Paul Klenerman |

The funders had no role in study design, data collection and interpretation, or the decision to submit the work for publication.

### Author contributions

Ghada Alsaleh, Conceptualization, Formal analysis, Validation, Investigation, Visualization, Methodology, Writing - original draft, Project administration, Writing - review and editing; Isabel Panse, Validation, Investigation, Methodology, set up the autophagy flux protocol in the primary cells and performed the experiments for the HCV vaccine samples and some of the RSV samples; Leo Swadling, Visualization, Writing - original draft; Hanlin Zhang, Investigation; Felix Clemens Richter, Methodology, Writing - review and editing; Alain Meyer, Eleanor Barnes, Resources; Janet Lord, Paul Klenerman, Christopher Green, Resources, Writing - original draft; Anna Katharina Simon, Conceptualization, Resources, Supervision, Funding acquisition, Visualization, Writing - original draft, Project administration, Writing - review and editing

## Author ORCIDs

Ghada Alsaleh (iD) https://orcid.org/0000-0002-4211-3420

Paul Klenerman (iD) https://orcid.org/0000-0003-4307-9161

## Ethics

Clinical trial registration NCT01070407 and NCT01296451.

Human subjects: Human vaccine samples: Human peripheral blood mononuclear cells (PBMC) were obtained under the ethics reference NRES Berkshire 13/SC/0023, from phase I clinical trials of novel viral-vectored vaccines for hepatitis-C virus (HCV; NCT01070407 and NCT01296451)or respiratory syncytial virus (RSV). Human healthy control: The study was approved by the Local Ethics Committee Oxford and Birmingham: The acquisition of normal control human tissue for medical research Kennedy Institute of Rheumatology. University of Oxford, Rec: 11/h0711/7 collection. University of Birmingham Research Ethics Committee, Reference ERN_12-1184R2, Investigations of the ageing immune system" Application for Ethical Review ERN_12-1184R2, UoB Ref: 17-1106.

Animal experimentation: Animal experiments were approved by the local ethical review committee and performed under UK project licenses PPL 30/3388.

## Decision letter and Author response

Decision letter https://doi.org/10.7554/eLife.57950.sa1

Author response https://doi.org/10.7554/eLife.57950.sa2

# Additional files

## Supplementary files

• Transparent reporting form

## Data availability

All data generated or analysed during this study are included in the manuscript and supporting files. Source data files have been provided for Figures 1-5 and figure supplements.

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
