## [Decision Letter]

**Acceptance summary:**

This paper involving autophagy and immune function is of high importance for the scientific community and public in general, especially in these times where the immune system is in a central focus. The paper now provide evidence that spermidine, in addition to its known healthful effects similar to those observed during caloric restriction including autophagy and protein deacetylation, it has the potential to become a new intervention for improving immune system health.

**Decision letter after peer review:**

Thank you for submitting your article "Autophagy in T cells from aged donors is maintained by spermidine, and correlates with function and vaccine responses" for consideration by *eLife*. Your article has been reviewed by two peer reviewers, and the evaluation has been overseen by a Reviewing Editor and Jessica Tyler as the Senior Editor. The reviewers have opted to remain anonymous.

The reviewers have discussed the reviews with one another and the Reviewing Editor has drafted this decision to help you prepare a revised submission.

Your article provides robust evidence that spermidine administration reinvigorates T-Cell function and autophagy in cells from older subjects. We believe that the following suggested revisions will greatly improve the clarify of your manuscript and thus, we hope you can address these promptly before we can accept it for publication. Some of the major conclusions require a modest amount of additional data as pointed by reviewer #2 and an extended Discussion as suggested by reviewer #3. Especially the proposed link between mitochondrial function and the improvement in autophagy is not supported by data. Please incorporate data on mitochondrial function or discuss prior published work. In addition, kindly provide evidence to demonstrate that spermidine effects on INF and T-cell function are blunted when autophagy is blocked, as suggested by reviewer #3.

Please find below the detailed suggestions from the reviewers.

Reviewer #2:

Simon and colleagues show that spermidine administration reinstalls autophagy and T-Cell function taken ex vivo from aged donors. They also point to hypusination and TFEB translation as an underlying mechanism. This is an intriguing and technically well done work and I have only a few suggestions for improvement.

The data showing that in young donors Autophagy is blocked by DFMO and can be rescued by spermidine administration are impressive and should be mentioned in the Abstract.

The sentence: “Here, we show induction of autophagy is specifically induced in human vaccine-induced antigen-specific T cells in vivo" is unclear and even flawed.

The authors correctly point out that mitochondria/mitophagy/improved mitochondrial function could play a role in the phenotypes observed. They should either show markers of mitochondrial capacity/quality in their assays (e.g. respiration, mitophagy) or at least discuss this in the light of published works.

Reviewer #3:

The manuscript entitled "Autophagy in T cells from aged donors is maintained by spermidine and correlates with function and vaccine responses" shows that antigen exposure induces autophagic flux in CD8^+^ T cells in humans in vivo, and this induction declines with age. Moreover, it evidences that the autophagy-inducing metabolite spermidine, falls with age and when supplemented increases autophagy and T cell function in old T cells.

The manuscript is overall well written and the message is potentially important. However, some conclusion are forced, and additional data are needed to support main author's conclusions.

– The correlation between INFɣ response and autophagic flux is positive in old, and negative in young individuals. The authors concluded that "reduced autophagy in aged T cells may cause low T cell response to vaccination". This is an interesting hypothesis that should be corroborated by direct evidences. For example, the authors should investigate INFɣ response in T cells in which autophagy was stimulated/inhibited using different compounds that modulate autophagy through different mechanism of action.

– The use of metabolite Spermidine has been previously shown to induce autophagy. However, the authors should include additional controls (i.e. other amino acid metabolites) to prove that the effects are indeed specific for spermidine.

– Spermidine's mode of action via EiF5A and TFEB in human CD8^+^ T cells requires additional experiments, i.e. real time of TFEB genes targets upon spermidine administration. Can the authors inhibit TFEB in spermidine treated cells to formally prove the involvement of TFEB in spermidine activity? Indeed, spermidine can activate autophagy through different mechanisms.

– There is no formal demonstration that Spermidine improve T-cell function through TFEB-mediated autophagy. The authors should use pharmacological inhibitors of autophagy to demonstrate that spermidine effects on INF and T-cell function are blunted when autophagy is blocked.

---

## [Author Response]

Reviewer #2:Simon and colleagues show that spermidine administration reinstalls autophagy and T-Cell function taken ex vivo from aged donors. They also point to hypusination and TFEB translation as an underlying mechanism. This is an intriguing and technically well done work and I have only a few suggestions for improvement.The data showing that in young donors Autophagy is blocked by DFMO and can be rescued by spermidine administration are impressive and should be mentioned in the Abstract.

We thank the reviewer for his/her recommendation and added a sentence describing this observation in the Abstract.

“We demonstrate in human donors that levels of the endogenous autophagy-inducing metabolite spermidine fall in T cells with age. Spermidine supplementation of T cells from old donors recovers their autophagy level and function, similar to young donor cells, in which spermidine biosynthesis has been inhibited.”

The sentence: “Here, we show induction of autophagy is specifically induced in human vaccine-induced antigen-specific T cells in vivo" is unclear and even flawed.We apologize for the unclear sentence and we are happy to clarify it. We replaced the sentence with the following in the Abstract:“Here, we show that autophagy is specifically induced in vaccine-induced antigen-specific CD8^+^ T cells in healthy human volunteers.”.The authors correctly point out that mitochondria/mitophagy/improved mitochondrial function could play a role in the phenotypes observed. They should either show markers of mitochondrial capacity/quality in their assays (e.g. respiration, mitophagy) or at least discuss this in the light of published works.

We appreciate the reviewer’s comment regarding the mitochondria. We addressed this point and added it to the Results, new Figure 3—figure supplement 2, Materials and methods and Discussion.

“Polyamine biosynthesis has previously been shown to increase mitochondrial protein synthesis (Madeo et al., 2018; Puleston et al., 2019; Wang et al., 2020). […] Mitochondria are not the only type of autophagic cargo and other cargos need to be investigated to understand exactly how autophagy induction improves T cell function.”

“Our data is in agreement with previous reports showing that spermidine increases the autophagy-dependent selective degradation of mitochondria (mitophagy) and thus contributes to mitochondrial health and functionality (Eisenberg et al., 2016; Garcia-Prat et al., 2016; Fan et al., 2017; Qi et al., 2016). The positive impact on cellular mitochondrial content is in line with two recent studies showing that TFEB promotes autophagy and improves mitophagy turnover in neuronal and hepatocytes (Tan et al., 2019; Kim et al., 2018).”.

“Mitochondrial staining was performed after Live/Dead and surface marker staining by incubating cells at 37 °C for 15 min with 5 µM MitoSOX red, 37 °C for 25 min with 150 nM MitoTracker Green, 37 °C for 25 min with 100 nM nonylacridine orange (NAO) or at 37 °C for 15 min with 100 nM tetramethylrhodamine methyl ester (TMRM) (all from Thermo Fisher Scientific).”

Reviewer #3:The manuscript entitled "Autophagy in T cells from aged donors is maintained by spermidine and correlates with function and vaccine responses" shows that antigen exposure induces autophagic flux in CD8^+^ T cells in humans in vivo, and this induction declines with age. Moreover, it evidences that the autophagy-inducing metabolite spermidine, falls with age and when supplemented increases autophagy and T cell function in old T cells.The manuscript is overall well written and the message is potentially important. However, some conclusion are forced, and additional data are needed to support main author's conclusions.– The correlation between INFɣ response and autophagic flux is positive in old, and negative in young individuals. The authors concluded that "reduced autophagy in aged T cells may cause low T cell response to vaccination". This is an interesting hypothesis that should be corroborated by direct evidences. For example, the authors should investigate INFɣ response in T cells in which autophagy was stimulated/inhibited using different compounds that modulate autophagy through different mechanism of action.

We appreciate the reviewer’s comment and we hope the newly added data provides additional information on the effects of decreased autophagy on T cell function.

To address this, we first used splenocytes from mice which were deleted for the essential autophagy gene Atg7 using the CD4Cre promotor. After stimulation, we found that autophagy loss leads to reduced IFNɣ, Perforin and Granzyme B expression in comparison to autophagy-sufficient cells (new Figure 2A-C). In addition, we treated old and young blood cells with either autophagy inhibitors (Hydroxychloroquine (HCQ)) and BSI-0206965 or with autophagy activators (Resveratrol) to assess their functional response. We performed intracellular cytokine staining to quantify IFNɣ, Perforin and Granzyme B response and ELISA for soluble IFNɣ in the supernatant (new Figure 2D-I). Our data shows that autophagy is important for T cell function. This was added to the Results, Materials and methods and new Figure 2.

“To gain better insight into the role of autophagy in T cells function, we sorted and activated splenocytes from control and autophagy-deficient mice, in which the essential autophagy gene Atg7 was deleted with Cre driven by the CD4 promotor (Atg7^Δcd4^). […] In summary, these data suggest that autophagy is required for CD8^+^ T cell function by controlling IFNγ and perforin production.”

“Mice T cell assays

Splenocytes from CD4^-cre^;Atg7^+/+^ and CD4^-cre^;Atg7^-/-^ mice were cultured with anti-CD3/CD28 for 4 days. For 4-Hydroxytamoxifen (4-OHT) 4-OHT- inducible Tfeb knockout splenic T cells (CAG-Cre/Esr1+, Tfebf/f), cells were stimulated were anti-CD3/CD28 and 100 nM 4-Hydroxytamoxifen (4-OHT) for 4 days.”

– The use of metabolite Spermidine has been previously shown to induce autophagy. However, the authors should include additional controls (i.e. other amino acid metabolites) to prove that the effects are indeed specific for spermidine.

Based on the reviewer’s suggestion, we treated the cells with spermine, another amino acid metabolite from the same pathway. Spermine was unable to rescue T cell function from old donors (new Figure 3—figure supplement 1). This was added to the Results, Materials and methods and new Figure 3—figure supplement 1 and is in line with our finding that spermine is not reduced with age (Figure 3A) and does not serve as substrate for eIF5A hypusination (Lee and Park, 2000).

“In contrast, spermine, another polyamine from the same pathway, which is not significantly reduced with age (Figure 3A) and does not serve as a direct substrate for eIF5A hypusination (Lee and Park, 2000), does not rescue CD8^+^ T cell function from old donors (Figure 3—figure supplement 1A-C). These data underline the importance of cellular spermidine levels in aged CD8^+^ T cells.”

– Spermidine's mode of action via EiF5A and TFEB in human CD8^+^ T cells requires additional experiments, i.e. real time of TFEB genes targets upon spermidine administration. Can the authors inhibit TFEB in spermidine treated cells to formally prove the involvement of TFEB in spermidine activity? Indeed, spermidine can activate autophagy through different mechanisms.

We appreciate the reviewer’s comment addressing the question of spermidine’s mode of action acting via TFEB. In an attempt to address this comment, we tested the Amaxa nucleofector kit optimised for human T cells. Unfortunately, transfection efficiency for plasmids expressing GFP was just under 16%, and no significant reduction of Tfeb mRNA was observed when primary human T cells were transfected with siTfeb for 3 days.

Instead, we obtained inducible Tfeb knockout splenic T cells from Andrea Ballabio’s lab. We tested several culture conditions and found an optimal protocol for inducing Tfeb knockout in these T cells ex vivo. Upon 100 nM 4-Hydroxytamoxifen (4-OHT) treatment for 4 days, T cells stimulated with anti-CD3/CD28 achieved a good inhibition (new Figure 5—figure supplement 2A).Functionally, IFNsecretionwassignificantlyreducedasmeasured in culture supernatants with or without spermidine (new Figure 5—figure supplement 2B). Taken together, these data suggest that TFEB is required for T cell function and that spermidine supplementation did not improve T cell function in this condition. This was added to the Results, Materials and methods and new Figure 5—figure supplement 2.

“To investigate if TFEB is required for T cell function, we knocked out Tfeb in cultured mouse T cells ex vivo (Figure 5—figure supplement 2A). […] Taken together, these data suggest that TFEB itself is essential for T cell function and that spermidine was unable to improve T cell function when TFEB was knocked out.”“Mice T cell assays

Splenocytes from: CD4^-cre^;Atg7^-/-^ (Atg7^Δcd4^) or CD4^-cre^;Atg7^+/+^ mice were cultured with anti-CD3/CD28 for 4 days. For 4-Hydroxytamoxifen (4-OHT) 4-OHT- inducible Tfeb knockout splenic T cells (CAG-Cre/Esr1+,Tfebf/f), cells were stimulated were anti-CD3/CD28 and 100 nM 4-Hydroxytamoxifen (4-OHT) for 4 days.”

– There is no formal demonstration that Spermidine improve T-cell function through TFEB-mediated autophagy. The authors should use pharmacological inhibitors of autophagy to demonstrate that spermidine effects on INF and T-cell function are blunted when autophagy is blocked.

We want to thank the reviewer for this comment and have added the requested experiment to the manuscript in the new Figure 3 with text in the Results and Materials and methods. In this figure we show that inhibition of autophagy using either HcQ or Sbl blocks spermidine-induced production of IFN**γ** measured in the supernatant or in each cell by flow cytometry. Interestingly, we observe similar changes for Perforin, whereas Granzyme B remains unchanged, similar to what we observed for the other assays. This data suggests the significance of autophagy in T cell function and cytokine secretion.

“Moreover, to investigate whether spermidine acts through autophagy to improve CD8^+^ T cell function, we added spermidine together with pharmacological autophagy inhibitors. Indeed, HcQ or Sbl ablates the rejuvenating effects of spermidine supplementation in old donor cells (Figure 3C-F).”